



# Measurement report: Structure of the atmospheric boundary layer and its relationship with the land-atmosphere interaction on the Tibetan Plateau

Maoshan Li[11], Wei Fu[2], Na Chang[1], Ming Gong[1], Pei Xu[1], Yaoming Ma [3], Zeyong Hu[4], Yaoxian Yang[4], Fanglin Sun [4]

(1.School of Atmospheric Sciences/Plateau Atmosphere and Environment Key Laboratory of Sichuan Province/Joint Laboratory of Climate and Environment Change, Chengdu University of Information Technology, Chengdu 610225, Sichuan China;

2. Yaan Meteorological Observatory, Sichuan Meteorological Bureau, 625000, Yaan, China)

3.Key Laboratory of Tibetan Environment Changes and Land Surface Processes, Institute of Tibetan Plateau Research, Chinese Academy of Sciences, CAS Center for Excellence in Tibetan Plateau Earth Sciences, Beijing, China;

4. Key Laboratory of Land Surface Process and Climate Change in Cold and Arid Regions, Chinese Academy of Sciences. Lanzhou, China)

**Abstract**

There is a deep atmospheric boundary layer on the Tibetan Plateau (TP) that has always been of interest to researchers. The variation in the atmospheric boundary layer under the influence of the southern branch of the westerly wind and that of the Asian monsoon was analyzed using sounding data collected in 2014 and 2019. Then, the hourly high-resolution comprehensive observation data for the land-atmosphere interaction on the TP and the ERA5 reanalysis data were used to study the influence of the atmospheric boundary layer's structure in Mount Everest, Nyingchi, Nam Co, Nagqu, and Shiquan River regions. The results show that the height of the convective boundary layer observed at the Mount Everest, Nyingchi, Nam Co, Nagqu, and Shiquan River stations on the TP under the influence of the southern branch of the westerly wind was higher than that during the Asian monsoon season. The height of the convective boundary layer in the Shiquan River area was often highest at 20:00. The structure of the boundary layer in

---

[1] Corresponding author

Dr. Maoshan Li

Chengdu University of Information Technology

Block 1, Xuefu Road, Chengdu 610225, Sichuan, China

E-mail: lims@cuit.edu.cn



the Mount Everest area was often affected by the westerly jets and glacial winds. The inversion
layer developed earlier in the Nyingchi area than at the other stations. The height of the boundary
layer was positively correlated with the sensible heat flux and negatively correlated with the
latent heat flux. The vertical velocity in the atmospheric boundary layer in the Nyingchi area
decreased, which may be one of the reasons why the height of the convective boundary layer was
lower in this area than at the other stations and humidity inversion often occurred in this area.
**Keywords: surface heat fluxes, structure of the atmospheric boundary layer, vertical**
**velocity, the southern branch of westerly wind, Asian monsoon**
**1 Introduction**

The Tibetan Plateau (TP), located in western China, is the highest plateau in the world. As a

high-elevation heat source up to the middle of the troposphere, the plateau can change matter and
energy directly in the middle and upper troposphere, and its thermal and dynamic effects have
important impacts on China's climate, the Asian monsoon, and even the global climate (Ye and
Gao, 1979; Chen et al., 1985; Yanai et al., 1992; Ye and Wu, 1998; Zhao and Chen, 2000; Wu et
al., 2012; Duan et al., 2013; Duan et al., 2017). The thermal effect is primarily manifested as a
heat source in summer and a heat sink in winter. As one of several subsystems of the Asian
monsoon system, the plateau monsoon is essentially an independent wind system formed under
the thermal activity on the plateau, and its characteristics are the most significant at 600 hPa.
Zheng and Wu (Zheng and Wu, 1995) showed that the northward jump of the southern branch of
the west wind in early summer is related to the thermal forcing on the plateau. The dynamic
effect of the plateau is mainly manifested in the obvious branching of the westerly wind. The
terrain blocking, and friction provided by the plateau itself forces the airflow to bypass this
region and climb, and in winter and summer, the flow around the area is greater than that of the
climbing air (Ye and Gao, 1979). The thermal and dynamic effects of the plateau on the
atmosphere affect the free atmosphere through the atmospheric boundary layer, so studying the
effects of the plateau boundary layer process is essential (Zhang and Hu, 2001).

The atmospheric boundary layer plays a vital role in regulating the energy and matter

transport from the Earth's surface into the free atmosphere (Zhang, 2003). Among them, the
height of the atmospheric boundary layer is an important index used to analyze the turbulent



mixing, vertical disturbance, convective transmission, and cloud band formation (Liu and Liang,
2010; Teixeira, et al., 2008). The height of the atmospheric boundary layer is related to the solar
radiation, type of underlying surface, weather systems, and topographic characteristics. When the
near-ground temperature is high, and the humidity is low, the surface sensible heat flux is
dominant, and the turbulence is enhanced, resulting in an increase in the height of the
atmospheric boundary layer (Zhang, et al., 2013). Many researchers have conducted in-depth
studies on the characteristics of the atmospheric boundary layer in different regions. Whiteman et
al. (Whiteman, et al., 2000) found that the rapid cooling of the atmosphere on the Mexican
plateau in the evening promotes a rapid change in the boundary layer on the plateau. Marshall et
al. (Marham, et al., 2008) observed a very thick convective boundary layer at up to 5500 m over
the Sahara Desert. Wang et al. (Wang, et al., 2016) found that the convective boundary layer over
the hinterland of the Taklimakan desert is deeply developed in summer, with a maximum height
of 4000 m. Through a field experiment conducted in the Heihe River Basin, Hu and Gao (Hu and
Gao, 1994) found that the sensible heat flux is the main component of the surface heat balance in
arid areas, and humidity inversion occurred in the atmosphere above oases and their adjacent
desert areas. Zhang et al. (Zhang, 1998; Zhang and Cao, 2003; Zhang, 2007) conducted field
observation experiments in the Gobi Desert in Dunhuang, northwest China in 2000. They found
that the thickness of the convective boundary layer on sunny days in summer exceeds 4000 m;
and under the arid climate, the energy consumed by the development of the atmospheric
boundary layer is much larger than that during its decline. Miao et al. (Miao, et al., 1998) found
that the complex terrain of the plateau enhances the mechanical turbulence movement, and the
height of the plateau boundary layer is higher than that of the plain area. Xu et al. (Xu et al.,
2001) comprehensively analyzed the plateau earth atmosphere physical process and dynamic
model based on the radiosonde observation data of the second Tibetan Plateau Atmospheric
Science Experiment in Chamdo, Damxung and Gerze. It is found that the thermal structure of the
Plateau Atmospheric boundary layer is abnormal, the development of the convective boundary
layer is deep, and the dynamic mechanism of Ekman "suction pump" in the plateau boundary
layer. Li et al. (Li, et al., 2000) used the observation data of the Tibetan Plateau Atmospheric
Science Experiments (TIPEX) Gerze station to find that there are many extreme values of wind
speed in the boundary layer of Gerze area and often inverse humidity. Zuo et al. (Zuo, et al.,





2004) comprehensively analyzed the observation data during the strengthening period in 1998. In
the dry season, the maximum height of the atmospheric boundary layer can reach 3550 m, and
the height of the boundary layer in the wet season is less than 2300 m. Su et al. (Su, et al., 2018)
found that the overall atmospheric boundary layer height on the plateau decreases significantly in
summer, the latent heat flux increases significantly, and the sensible heat flux initially increases
and then decreases. Zhou et al. (Zhou et al., 2018) found that the height of the plateau boundary
layer is high in the west and low in the east using cosmic occultation data. The height of the
western boundary layer is 1800-2300 m, and the height of the eastern boundary layer is
1400-1800 m.

Compared with the plain area, the land-atmosphere interaction on the TP is intense and

complex, which affects the structure of the boundary layer. The large temperature difference
between the land and the atmosphere, the strong solar radiation on the ground, and its complex
topographic characteristics results in the boundary layer above the plateau having a unique and
complex structure; thus, it is particularly important to study the changes and characteristics of the
atmospheric boundary layer on the TP. In light of this, the characteristics of the boundary layer in
different regions of the plateau were analyzed in this study. Then, the relationships between the
boundary layer's structure and the sensible heat flux, latent heat flux, and vertical velocity field
were studied to gain a deeper understanding of the variations in the height of the atmospheric
boundary layer above the plateau, the specific humidity, wind direction, and wind speed under
the influence of the southern branch of the westerly wind as well as the Asian monsoon, to
explore the mechanism of the variations in the boundary layer, and provide a theoretical basis for
the prediction and early warning of plateau weather and climate under the background of climate
change in the future.
**2. Data, study area, and methods**
**2.1 Data description and study area**

(1) Sounding data collected at the Mount Everest, Nyingchi, and Namco stations in 2014

and sounding observations collected at the Shiquan River station during the Earth-atmosphere
Interaction and Climate Effect Enhanced Observation Experiment of the Second Tibetan Plateau
Scientific Expedition in 2019 were used in this study. The locations of and basic information
about the five sounding stations are presented in Figure 1 and Table 1, respectively.





All data times used in this study were Beijing standard time (BST, BST= Coordinated
Universal Time (UTC)+8). In 2014, the daily sounding observation time of the three stations of
Mount Everest, Nyingchi and Shiquan River was 08:00, 14:00 and 20:00. In 2019, the daily
sounding observation time of the four stations of Mount Everest, Nyingchi, Nagqu and Shiquan
River was 02:00, 08:00, 14:00 and 20:00. The sounding observation includes meteorological
elements such as temperature, air pressure, humidity, wind speed and wind direction, and the
data collection frequency is 2 s.
(2)ERA5 reanalyzes UV wind speed, surface sensible heat, latent heat flux, and vertical
velocity data (https://www.ecmwf.int/en/about/ media-centre/ science-blog/2017/era5-new
-reanalysis-weather-and-climate-data), with a horizontal resolution of 0.25°×0.25°.
(3)A long-term dataset of integrated land-atmosphere interaction observations on the
Tibetan Plateau (2005-2016) (Ma et al., 2020). Hour-by-hour sensible heat and latent heat flux
turbulence data from the Qomolangma Atmospheric and Environmental Comprehensive
Observation and Research Station, the Southeastern Tibet Alpine Environment Comprehensive
Observation and Research Station, and the Namco Multi-Circle Comprehensive Observation and
Research Station in 2014 were used. In this paper, sensible heat and latent heat flux data with
data quality status quality assurance less than 4 are selected for research.

**Figure 1 about here**
**Table 1 about here**
**2.2 Methods**
Regarding the methods for determining the height of the boundary layer, there are many
methods such as the potential temperature gradient method, the Holzworth method (dry adiabatic
method, the gas block method), the Richardson method, and the potential temperature profile
method (Sullivan, et al.,1997; Seibert, et al., 2000; Seidel, et al., 2010; Liu, et al., 2013; Stefan and
Klaus, 2006). In this paper, the determination of the boundary layer height was mainly based on
the potential temperature profile method, and the height with an obvious discontinuous gradient
was regarded as the height of the atmospheric boundary layer. During the day, the height of the
capping with strong inversion is taken as the height of the convective boundary layer. At night, the
height of the ground inversion layer was taken as the height of the stable boundary layer. From the
perspective of dynamic action, the height of the stable boundary layer at night can be regarded as
the height of the stable boundary layer according to the height of the maximum wind speed. At the
same time, from the perspective of humidity distribution, the height of the convective boundary
layer can be regarded as the height where the specific humidity decreased rapidly. Unless
otherwise specified, the heights used in the text are relative heights.
**3 Structure of the atmospheric boundary layer under the influence of the southern branch**



**of the westerly wind as well as the Asian monsoon**
Due to the particularity of the plateau geography and climate, the observational
understanding of the atmospheric boundary layer structure in plateau areas is still limited
compared with the observational understanding of the atmospheric boundary layer structure in
other regions. This paper uses the sounding data between 2014 and 2019 to study the change
characteristics of vertical meteorological elements in the plateau area. The focus is on the diurnal
variation characteristics of the atmospheric boundary layer height, potential temperature profile,
specific humidity profile, and wind speed and direction profile in the coordinated area of the
south branch of the westerly wind and Asian monsoon.
**3.1 Structure of the atmospheric boundary layer under the influence of the southern**
**branch of the westerly wind as well as the Asian monsoon**
**3.1.1 Height of the convective boundary layer under the influence of the southern branch of**
**the westerly wind as well as the Asian monsoon**
Figure 2 shows the potential temperature profiles in the convective boundary layer observed
at the Mount Everest, Nyingchi, and Nam Co stations in June, August, and November 2014 and at
the Shiquan River station in May, July, and October 2019. Because the low-level interval of the
radiosonde data collected at the Nyingchi and Nam Co stations is large, the potential temperature
profile in the lower level is discontinuous. However, it can still be seen that the potential
temperature profile is in a convective state at the lower level, and it does not affect the
determination of the height of the atmospheric boundary layer. Under the influence of the plateau
monsoon in June, the heights of the mixed layer were 3000 m, 2100 m, 2200 m, and 2650 m at
Mount Everest, Nyingchi, Nam Co, and Shiquan River stations, respectively. Under the influence
of the plateau summer monsoon wind field in August, the heights of the mixed layer were 1700 m,
1000 m, 950 m, and 2000 m at Mount Everest, Nyingchi, Nam Co, and Shiquan River stations,
respectively. Under the influence of the southern branch of the westerly wind field in November,
the heights of the mixed layer were 4500 m, 3000 m, 2400 m, and 3,500 m at Mount Everest,
Nyingchi, Nam Co, and Shiquan River stations, respectively.
In summary, the height of the mixed layer exhibited distinct characteristics at the Mount
Everest, Nyingchi, Nam Co, and Shiquan River stations under the influences of the different
circulation background fields. At all of the stations, the height of the mixed layer under the
influence of the southern branch of the westerly wind field (May, October, and November) was
higher than that under the influence of the Asian monsoon (June, August, and July).



**Figure 2 about here**

**3.1.2 Diurnal variations in potential temperature**

Figure 3 presents the potential temperature profiles for the Mount Everest, Nyingchi, and

Nam Co stations in June, August, November 2014 and the Shiquan River station in May, July, and
October 2019. At 08:00 on June 8, at the Mount Everest station, the height of the stable boundary
layer was about 240 m. The height of the convective boundary layer had increased to 3000 m by
14:00, and the upper caping temperature inversion was located at 3000–3350 m. At 20:00, the
height of the convective boundary layer was 2900 m, and there was weak inversion stratification
between 1400 and 1500 m and between 2250 and 2300 m, which may have been affected by the
westerly jet with wind speeds of up to 16 m s$^{-1}$ between 1500–2000 m. At 08:00 on August 26, the
height of the stable boundary layer was about 100 m. At 14:00, the atmosphere was further heated
by surface heating, and the height of the convective boundary layer increased to 750 m. At 20:00,
there was an obvious super-insulation layer in the lower layer of the potential temperature profile,
the lower atmosphere stratification was statically unstable, and the height of the convective
boundary layer reached 1600 m. At 08:00 on November 23, the height of the stable boundary layer
was about 750 m. At 14:00, the height of the convective boundary layer reached 4500 m. At 20:00,
the height of the convective boundary layer was 3700 m, and the temperature inversion of the
upper capping layer occurred between 3700 and 3800 m.

On June 12, at the Nyingchi station, it was impossible to accurately determine the height of

the atmospheric boundary layer at 08:00 due to the large interval between the sounding data for
the lower layers. At 14:00, there was an obvious super-insulation layer in the lower layer of the
potential temperature profile, the stratification of the lower atmosphere was unstable, and the
height of the convective boundary layer was 2100 m. At 20:00, a stable boundary layer began to
develop. On August 24, due to the large interval between the sounding data for the lower layers,
the height of the atmospheric boundary layer could not be accurately determined. However,
according to the potential temperature profiles at 14:00 and 20:00, the height of the atmospheric
boundary layer should not have been higher than 1000 m. At 08:00 on November 25, the height of
the stable boundary layer was 1500 m, and the residual layer was located at 1500–2800 m. At
14:00, the height of the convective boundary layer was 3000 m. The stable layer began to develop
at 20:00, and the residual layer was located at 300–3200 m. At 08:00 on June 9, the surface
heating destroyed the stable boundary layer at night, and convection began to develop. At 14:00,
the height of the convective mixed layer was about 2200 m. At 20:00, the height of the convective





mixing layer was 1600 m. On August 26, due to the large interval between the sounding data for
the lower layers, it was impossible to accurately determine the height of the atmospheric boundary
layer. However, according to the potential temperature profiles at 14:00 and 20:00, the height of
the boundary layer was 1000 m. At 08:00 on November 29, the height of the stable boundary layer
was 150 m. At 14:00, the height of the convective boundary layer was 2400 m.

The heights of the stable boundary layer at the Shiquan River station were 250 m and 150 m

at 2:00 and 8:00 on May 16, respectively. The heights of the convective boundary layer were 2550
m and 2650 m at 14:00 and 20:00, respectively. The height of the stable boundary layer at the
Shiquan River station was 500 m at 02:00 on the night of July 28. The heights of the convective
boundary layer were 1350 m and 1700 m at 14:00 and 20:00, respectively. The height of the stable
boundary layer at the Shiquan River station was 100 m at 02:00 on the night of October 23, and
the height of the stable boundary layer was 150 m at 08:00. The heights of the convective
boundary layer were 2150 m and 3500 m at 14:00 and 20:00, respectively.

**Figure 3 about here**


The above examples for each station illustrate that the changes in the atmospheric boundary

height above the different areas of the plateau under different background wind fields. The height
of the atmospheric boundary layer at each station had obvious diurnal characteristics. The
structure of the boundary layer in the Mount Everest area was often affected by the westerly jets
and glacial winds, resulting in a more complex potential temperature profile and a special
boundary layer structure with a lower boundary layer. The inversion layer developed earlier in
the Nyingchi area than at the other stations, and it began to develop near the low-level layer at
20:00. The height of the convective boundary layer in the Shiquan River area often reached the
highest point at 20:00.
**3.1.3 Vertical distribution of specific humidity**

Figure 4 shows that the near-surface layer specific humidity values at the Everest, Nyingchi,

and Nam Co stations in November were much lower than in June and August in 2014. In August
and November, the specific humidity of the surface layer was larger at the Nyingchi station than at
the Mount Everest and Nam Co stations. At the Nyingchi and Nam Co stations, due to the large
interval between the data for the lower level, the low-level specific humidity profile was not
smooth in November. The relative humidity of the surface layer at the Mount Everest, Nyingchi,
and Nam Co stations in June was quite similar to that of the surface layer in November. At each


station, the relative humidity of the surface layer was larger at 08:00 than at 14:00 and 20:00. On
November 23, 2014, the lower layer above the Mount Everest station was affected by the northerly
valley wind (wind speed of 20.9 m s$^{-1}$) at 14:00 and by the southerly glacial wind (wind speed of
15 m s$^{-1}$) at 20:00, and humidity inversion occurred at both times. The specific humidity of the
surface layer at the Shiquan River station (Fig. 4d) followed the order of July > May > October.
The humidity of the surface layer was lower at the Shiquan River station than are the Nyingchi,
Nagqu, and Nam Co stations in May and October. The humidity of the surface layer was lower at
the Mount Everest station than at the Nyingchi, Nam Co, and Shiquan River stations in July. The
specific humidity of the near-surface layer was significantly higher at the Nyingchi station than at
the other stations, and the maximum specific humidity was $12.88\,\mathrm{g} \bullet \mathrm{kg}^{-1}$. Mount Everest was
affected by the low-level jet at 08:00 and the glacial wind, which had higher wind speeds, at 16:00
and 20:00 on October 29, 2019. The specific humidity of the lower level also exhibited humidity
inversion.
The lower layer-specific humidity of the atmospheric boundary layer exhibited obvious
diurnal variations at all the stations (Fig. 4). The near-surface layer had a higher humidity at night
than during the day, and the specific humidity decreased as the height increased. The phenomenon
of humidity inversion often occurred at the Mount Everest and Nyingchi stations. Temperature
inversion occurred in the stable boundary layer and near the top of the convective boundary layer,
while the transport of water vapor by the low-level jets and the existence of downdrafts resulted in
humidity inversion in the near-surface layer. The specific humidity of the lower atmospheric
boundary layer at each station was lower under the influence of the southern branch of the
westerly wind field than under the influence of the Asian monsoon wind field.

**Figure 4 about here**

**3.1.4 Vertical distributions of wind speed and wind direction**
In addition to the potential temperature and specific humidity, wind is also one of the main
meteorological elements involved in energy and matter transport and exchange processes in the
atmospheric boundary layer. Figs. 5 and 6 present the wind speed and direction profiles,
respectively, for the Mount Everest, Nyingchi, and Nam Co stations in 2014 and the Shiquan
River station in 2019. On June 8, the wind speed near the surface at the Mount Everest station was
lower at 08:00 than at 14:00 and 20:00. At 14:00 on June 8 and August 26, the low-level wind
direction was north-northeast, which was affected by the valley wind and the northerly wind
before 14:30. This was discovered by Chen et al. (Chen, et al., 2007) during the rainy season on



Mount Everest. On June 8, the Mount Everest area was dominated by westerly winds above 1200
m. On August 26, it was dominated by northwest and west-northwest winds above 2000 m. On
November 25, it was dominated by westerly winds above 1000 m. The wind speed in the
near-surface layer in the Nyingchi area was lower at 08:00 than at 14:00 and 20:00, and the wind
speed in the near-surface layer was lower in the Nyingchi area than in the Mount Everest and Nam
Co areas. On June 12, the southerly wind prevailed in the Nyingchi area above 1000 m, but the
prevailing wind changed to the westerly wind above 2000 m. The westerly winds were dominant
above 3000 m on August 24, and the westerly winds were dominant above 4000 m on November
25. The wind speed in the Nam Co area was higher on November 29 than on June 9 and August 26.
The wind direction was from the south to the west above 1000 m on June 9 at the Shiquan River
station. On August 26 and November 29, the wind direction above the surface layer was westerly.
The wind direction was mainly west-southwest on May 16 and October 23, and the low-level jets
of the westerly wind appeared near the surface at 20:00 at the Shiquan River station. The wind
direction was mainly west-northwest on July 28, the low-level wind speed was much stronger on
May 16 and October 23 than on July 28. The wind speed in the upper air at the Mount Everest,
Nyingchi, and Shiquan River stations was stronger in May and October than in July, and the wind
direction was mainly westerly or southwesterly.
The wind speed exhibited obvious diurnal characteristics and increased with increasing
height at all of the stations. They were also significantly different at each station. The wind speed
and direction were often affected by the valley wind or glacier wind at the Mount Everest station.
The Nyingchi station may have been affected by its geographical location and topography. The
low-level wind direction was mostly southerly, and the low-level wind speed was lower at the
Nyingchi station than at the Mount Everest, Nam Co, and Shiquan River stations. At all the
stations, the wind speed increased more rapidly with the height under the southern branch of the
circulation field of the westerly wind. The wind direction in the upper level may have been
primarily affected by the westerly wind. At all the stations, the wind direction and wind speed in
the lower level were greatly affected by the topography and geographical location, while the
changes in the wind direction and wind speed in the upper level were related to the large-scale
westerly circumfluence and the Asian monsoon.

**Figure 5 about here**

**Figure 6 about here**

**4 Variations in the surface energy fluxes and the structure of the atmospheric boundary**
**layer under the influence of the south branch of the westerly wind as well as Asian monsoon**





**on the TP**
The height and structure of the atmospheric boundary layer have great temporal and spatial
differences, and the thermal effect of the land surface is one of the important reasons for the
formation and change of the atmospheric boundary layer. Sensible heat flux is the heat transfer
between the ground and the atmosphere caused by turbulent motion in the near-surface layer, and
latent heat flux is the heat transfer between the underlying surface and the atmosphere caused by
the phase change of water in the atmosphere. Su et al. showed that the height of the atmospheric
boundary layer in the plateau is generally positively correlated with the surface sensible heat flux,
and negatively correlated with the surface latent heat flux (Su, et al., 2018). This study intends to
use the observational data and ERA5 reanalysis data to understand the regional differences in
sensible and latent heat fluxes, analyze the differences in the characteristics of sensible heat and
latent heat fluxes under the coordinated action of the westerly wind and monsoon, and analyze the
relationship between the height of the atmospheric boundary layer and the energy heat fluxes.
**4.1 Comparison of the sensible and latent heat fluxes between in-situ observations and ERA5**
**reanalysis data**
The sensible and latent heat flux from the ERA5 reanalysis dataset were evaluated using the
observations at Mount Everest, Nyingchi, and Nam Co stations on June 4–12, August 20–29
(Asian monsoon), and November 21–30 (westerly south branch) in 2014. Figure 7 presents a
comparison of the observed sensible and latent heat fluxes and the ERA5 sensible and latent heat
fluxes during the sounding period at the Mount Everest, Nyingchi, and Nam Co stations in 2014.
Nyingchi lacks observational sensible and latent heat flux data for June and August 2014. The
observational heat flux data for each station and the ERA5 reanalysis data exhibited significant
diurnal changes. The turbulence intensity was weak, the sensible and latent heat fluxes were small
under the stable atmospheric stratification conditions at night, and the atmosphere transferred heat
down to the surface when the sensible heat flux decreased to negative values. After sunrise, the
solar radiation heated the surface, the sensible and latent heat fluxes gradually increased, and the
turbulence strengthened. Through comparison of the two sensible and latent heat flux datasets in
June, August, and November, it was found that the observed sensible heat flux was greater than
the ERA5 sensible heat flux, and the ERA5 latent heat flux was greater than the observed latent
heat flux. Excluding the observation data in June, when the sensible heat flux was greater than the
latent heat flux, the latent heat flux of the ERA5 reanalysis data was greater than the sensible heat
flux. The comparison of the August observation data and the ERA5 reanalysis data also shows that
the latent heat flux was greater than the sensible heat flux. Both the observation data and the
ERA5 reanalysis data show that the latent heat flux was greater than the sensible heat flux in





November. At the Mount Everest, Nyingchi, and Nam Co stations, the latent heat flux observed in
August > the latent heat flux observed in June > the latent heat flux observed in November and the
sensible heat flux observed in June > the sensible heat flux observed in November > the sensible
heat flux observed in August. According to the above results for the height of the convective
boundary layer in November and August, the height of the boundary layer was positively
correlated with sensible heat flux and negatively correlated with the latent heat flux. This is
consistent with the results of Sun et al. (Sun, et al., 2021) on the height of the atmospheric
boundary layer and the sensible and latent heat fluxes on the plateau.
Figure 8 shows a comparison of the sensible and latent heat flux observations at the Everest,
Nyingchi, and Nam Co stations in June, August, and November and the ERA5 reanalysis data.
The correlation coefficients between the ERA5 reanalysis sensible heat flux data and the
measured data for the Mount Everest, Nyingchi, and Nam Co stations are 0.85, 0.74, and 0.77,
respectively. The correlation coefficients between the ERA5 reanalysis latent heat flux data and
the measured data at the Mount Everest, Nyingchi, and Nam Co stations are 0.5, 0.52, and 0.42,
respectively. Compared with the latent heat flux, the sensible heat flux ERA5 data have better
applicability in the different plateau areas. The correlation coefficients between the ERA5 and
the measured sensible heat flux data are consistent with the results of Zhu et al.'s evaluation of
the correlation coefficients between ERA40 and measured data (Zhu, et al., 2012). The observed
sensible and latent heat fluxes and the ERA5 sensible and latent heat flux data exhibited certain
differences during the observation period, and these differences were mainly due to the
difference in their spatial resolutions. The correlation coefficients between the observed and
ERA5 latent heat flux data at the Mount Everest, Nyingchi, and Nam Co stations are relatively
small. This is because, on the same day, the weather conditions of the two datasets are likely to
be different. This may be due to the influence of rainfall during this period. This is consistent
with the research results of Sun et al. (Sun, et al., 2021). In summary, the consistency between
the ERA5 reanalysis and measured sensible and latent heat flux datasets is still good and can be
used to study the characteristics of the surface energy in the plateau areas.
**Figure 7 about here**
**Figure 8 about here**
**4.2 Effect of land surface heating on the atmospheric boundary layer**
The height and structure of the atmospheric boundary layer vary temporally and spatially.
The atmosphere-land interaction is one of the important reasons for the formation of and changes
in the atmospheric boundary layer. Su et al. showed that the height of the atmospheric boundary
layer above the plateau is generally positively correlated with the surface sensible heat flux, while



it is negatively correlated with the surface latent heat flux (Su, et al., 2018). In this study,
observational and ERA5 reanalysis data were used to gain a better understanding of the regional
differences in the sensible and latent heat fluxes, to analyze the differences in changes in the
sensible and latent heat fluxes under the combined action of the westerly wind and monsoon, and
to explore the mechanism of the relationships between the height of the atmospheric boundary
layer and the sensible and latent heat flux.
**4.2.1 Relationships between the atmospheric boundary layer and the sensible and latent**
**heat fluxes**
Figure 9 shows a comparison of the measured and ERA5 reanalysis sensible and latent heat
flux data, at the Everest, Nyingchi, NagquShiquan River stations in May, October (under the
influence of the southern branch of the westerly wind field), and July (under the plateau summer
wind field) in 2019. The sensible and latent heat fluxes exhibited obvious diurnal variations at all
of the stations. Beginning in the morning, solar radiation heated the ground, and the heat was
transferred from the surface into the atmosphere in the form of sensible and latent heat fluxes. The
sensible and latent heat fluxes reached their peaks at noon, and then, they gradually decreased.
When they decreased to a negative value, the heat was transferred from the atmosphere to the
surface. The sensible heat flux was relatively large in May and October at the Mount Everest and
Shiquan River stations, while the latent heat flux was relatively small, which is consistent with the
fact that the boundary layer heights were higher at the Mount Everest and Shiquan River stations.
The sensible and latent heat fluxes at the Nyingchi and Nagqu stations were similar in May and
October, which is consistent with the results of Zheng et al. (Zheng, et al., 2019). This may be one
of the reasons that the height of the boundary layer was lower at the Nyingchi and Nagqu stations
than at the Mount Everest and Shiquan River stations. In July, the latent heat flux was significantly
higher than the sensible heat flux at the Mount Everest, Nyingchi, Nagqu, and Shiquan River
stations. This was mainly because summer included a period of concentrated precipitation and a
period of vigorous vegetation growth. The sensible heat flux was greater than the latent heat flux
at all the stations under the influence of the southern branch of the westerly wind. The
near-surface energy exchange was dominated by the sensible heat flux. The boundary layer
experienced strong atmospheric turbulence and strong convection, which increased the height of
the boundary layer. However, the latent heat flux was larger at all the stations under the influence
of the Asian summer monsoon wind field, and the water vapor content of the air was higher, which
inhibited the development of the boundary layer.

**Figure 9 about here**

**Table 2 about here**





**Table 3 about here**
Tables 2 and 3 present the convective boundary layer heights and the maximum sensible and
latent heat fluxes at each station during each sounding period in 2014 and 2019. Due to the
differences between the observed fluxes and the ERA5 reanalysis fluxes, the analysis in Table 2
mainly focused on the observed fluxes. As can be seen from Table 2, at all the stations, the height
of the boundary layer was the lowest in August, and the corresponding sensible heat flux values
were lower than those in June and November, while the latent heat flux values were much higher
than those in November. As can be seen from Table 3, at all the stations, the height of the
boundary layer was the lowest in July, and the corresponding latent heat fluxes were much higher
than those in May and October. Therefore, based on the analysis of the height of the boundary
layer and the sensible and latent heat fluxes at each station, the height of the boundary layer was
positively correlated with the sensible heat flux and negatively correlated with the latent heat flux.
**4.2.2 Relationships between the structure of the atmospheric boundary layer and the**
**vertical velocity distribution**
Energy conversion in the atmosphere is mainly achieved through the vertical movement of
the atmosphere, which has a great influence on the transportation of water vapor, matter, and
energy. Figure 10 shows the variation in the vertical velocity in the convective boundary layer
during each radiosonde observation period at the Mount Everest, Nyingchi, Nam Co, Nagqu, and
Shiquan River stations. The vertical velocity exhibited obvious diurnal characteristics, and the
vertical velocity near the surface layer was mainly concentrated between -0.2 and 0.2 Pa s$^{-1}$ at
night, which is consistent with the results of Xu et al. (Xu, et al., 2006) and Cao et al. (Cao, et al.,
2017). The vertical velocities in the lower layers at the Mount Everest, Nam Co, Nagqu, and
Shiquan River stations were mostly indicative of updrafts; while strong downdrafts mainly
occurred in the lower layers at the Nyingchi station, which was not conducive to the development
of convection in the atmospheric boundary layer. Figures 10a, d, and g show the vertical velocity
profiles at the Mount Everest, Nyingchi, and Nam Co stations under the influence of the Asian
monsoon in June. There was only weak upward movement below 500 hPa during the daytime, and
strong downdrafts occurred after 15:00. There were strong downdrafts above 500 hPa at 18:00 at
the Nyingchi and Nam Co stations. At night, weak sinking movement occurred at the Mount
Everest station, strong sinking movement occurred at the Nyingchi station, and strong ascending
movement occurred at 400 hPa at the Nam Co station. Figures 10b, e, and h show the vertical
velocities at the Mount Everest, Nyingchi, and Nam Co stations under the influence of the Asian
monsoon field in August. The lower level weakly ascended from 09:00 to 21:00 at the Mount
Everest station. The lower layers contained sinking air almost all day at the Nyingchi station, but a





strong upward movement occurred above 500 hPa from 09:00 to 12:00. In the lower layers, weak
upward and downward movements alternately occurred at the Nam Co station. In 2014, Figures
10c, f, and i show the vertical velocity profiles at the Mount Everest, Nyingchi, and Nam Co
stations under the southern branch of the westerly wind field. At the Mount Everest station, the
lower layer exhibited ascending movement from 10:00 to 17:00, while strong sinking motion
centered at 400 hPa occurred at 0:00 and was gradually replaced by ascending motion at 200 hPa
over time. At the Nyingchi station, there was almost sinking movement throughout the day below
250 hPa, but the sinking movement was weaker during the day than at night. At the Nam Co
station, below 300 hPa, there was almost ascending movement throughout the day, and above 300
hPa there was sinking movement. At the Shiquan River station, strong vertical ascent occurred
after 18:00 (Figs. 10j, k, l). This may be due to the heating of the plateau area by solar radiation
during the afternoon, which increased the surface temperature. The energy was gradually
transferred upwards, the convection gradually strengthened, the boundary layer became fully
mixed, and the vertical velocity gradually increased. Figure 10 shows the vertical velocity profile
at the Shiquan River station under the influence of the southern branch of the westerly wind field.
At the Shiquan River station, the ascending movement gradually increased after 12:00.

**Figure 10 about here**

According to the analysis of Figure 10, convection was active in the boundary layer during
the day and ascending and sinking motions alternately occurred. During the day, as the solar
radiation increased, the accumulation of surface heat and the atmospheric turbulence increased,
and the vertical velocity mostly reached the maximum at various heights at 18:00 or later. At
14:00 on May 16 and October 25, 2019, a strong sinking movement occurred at the Nyingchi
station. In addition, humidity inversion occurred in the lower level. The vertical velocity in the
atmospheric boundary layer at the Nyingchi station was sinking. This may be one of the reasons
that the height of the convective boundary layer was lower at the Nyingchi station than at the
other stations, and it is also one of the reasons that humidity inversion frequently occurred at the
Nyingchi station.
**5 Discussion and conclusions**
The variations in the temperature, specific humidity, wind speed, and wind direction profiles
in the atmospheric boundary layer under different wind fields were analyzed using the sounding
data collected in June, August, and November in 2014 and in May, July, and October in 2019.
Then, the hourly high-resolution comprehensive observation data from the TP land-atmosphere



interactions and ERA5 reanalysis data were used to study the influences on the structure of the
atmospheric boundary layer at Mount Everest, Nyingchi, Nam Co, Nagqu, and Shiquan River
stations. The results are as follows.
(1) The heights of the convective boundary layer observed at the Mount Everest, Nyingchi,
Nam Co, Nagqu, and Shiquan River stations on the plateau under the influence of the southern
branch of the westerly wind were 4500 m, 3000 m, 2400 m, 2760 m, and 3500 m, respectively.
Under the influence of the Asian monsoon, the heights of the convective boundary layer observed
at Mount Everest, Nyingchi, Nam Co, Nagqu, and Shiquan River stations on the plateau were
3000 m, 2100 m, 2200 m, 1650 m, and 2000 m, respectively. At each station, the height of the
convective boundary layer was higher under the influence of the southern branch of the westerly
wind than under the influence of the Asian monsoon. The structure of the boundary layer at the
Mount Everest station was often affected by the westerly jets and glacial winds, resulting in a
more complex potential temperature profile and a special low boundary layer height. The
inversion developed earlier at the Nyingchi station than at the other stations, and it began to
develop near the low-level inversion layer at 20:00. The height of the convective boundary layer at
the Shiquan River station was often highest at 20:00.
(2) Under the influences of the southern branch of the westerly wind and the Asian monsoon,
the lower specific humidity of the atmospheric boundary layer at each station exhibited obvious
diurnal variations. Humidity inversion often occurred at the Mount Everest and Nyingchi stations.
The inversion layers, low-level jets to transport water vapor, and sinking airflow may all have
contributed to the occurrence of humidity inversion in the near-surface layer. The specific
humidity was lower in the lower layer of the atmospheric boundary layer under the influence of
the southern branch of the westerly wind than that under the Asian monsoon at all of the stations.
(3) The wind speed and direction at the Mount Everest station were often affected by the
valley winds and/or glacier winds, and jets often appeared in the lower layers. The atmospheric
structure in the Nyingchi area may be affected by its geographical location and topography. The
wind direction in the lower level was mostly southerly, and the wind speed in the lower level was
lower at the Nyingchi station than at the Mount Everest, Nam Co, and Shiquan River stations.
Under the influence of the southern branch of the westerly wind, at all of the stations, the wind
speed increased more rapidly with height, and the wind direction in the upper level may have been
affected by the westerly winds. The wind direction and wind speed in the lower level were greatly
affected by the topography and geographical location of each station, while the changes in the
wind direction and wind speed in the higher level were related to the large-scale westerly
circumfluence and the Asian monsoon.



(4) The diurnal variations in the heat fluxes were very significant at the Mount Everest, Nyingchi, Nam Co, Nagqu, and Shiquan River stations, exhibiting unimodal variations. In 2014, the observed latent heat flux in August > the observed latent heat flux in June > the observed latent heat flux in November; and the observed sensible heat flux in June > the observed sensible heat flux in November > the observed sensible heat flux in August at the Mount Everest, Nyingchi, and Nam Co stations. According to the height of the convective boundary layer in November and the fact that the height of the convective boundary layer was lowest in August, the height of the boundary layer was positively correlated with the sensible heat flux and negatively correlated with latent heat flux. In 2019, the sensible heat flux was relatively large in May and October at the Mount Everest and Shiquan River stations, while the latent heat flux was relatively small. The sensible heat flux and latent heat flux were similar in May and October at the Nyingchi and Nagqu stations, which corresponded to the occurrence of higher boundary layer heights at the Mount Everest and Shiquan River stations and low boundary layer heights at the Nyingchi and Nagqu stations. In July, the latent heat flux was larger than the sensible heat flux at the Mount Everest, Nyingchi, Nagqu, and Shiquan River stations. The main reason for this was that summer was a period of concentrated precipitation and a period of vigorous vegetation growth. The latent heat flux was small at all of the stations under the influence of the southern branch of the westerly wind, and the energy transport near the ground was dominated by the sensible heat flux. The boundary layer exhibited strong atmospheric turbulence and convection, which increased the height of the boundary layer. However, the latent heat flux and the moisture content in the atmosphere were high at all of the stations under the influence of the plateau summer monsoon wind field, which inhibited the development of the boundary layer. At 14:00 on May 16 and October 25, 2019, a strong sinking movement occurred at the Nyingchi station. In addition, humidity inversion occurred in the lower level. The vertical velocity in the atmospheric boundary layer at the Nyingchi station was sinking. This may be one of the reasons why the height of the convective boundary layer was lower at the Nyingchi station than at the other stations. This is also one of the reasons why humidity inversion frequently occurred at the Nyingchi station.

Due to its vast size and complex terrain, observation data for the TP are scarce. In conventional meteorological observations, not only is the observation history short but the precision of the observation instruments and the content of the observation parameters are also limited, which may affect the accuracy of the analysis. The sounding data used in this paper were limited in time and space, and more extensive spatial and long-term research and analysis are lacking. If sounding data with a higher temporal resolution were obtained, the structure of the





atmospheric boundary layer on the TP could be analyzed in more detail. In this study, only the
relationships between the structure of the boundary layer and the sensible heat flux, latent heat
flux, and vertical velocity on the TP were investigated, and the reasons for the structure of the
boundary layer in terms of the turbulence intensity were not analyzed. In addition, numerical
simulations of the atmospheric boundary layer's structure can be carried out based on the
structure of the atmospheric boundary layer obtained from the sounding data. These issues are
also topics that can be studied further.

**Author Contributions:** M.L., W.F., N.C. and Y.Y. mainly wrote the manuscript and were
respon-sible for the research design, data preparation and analysis. Y.M., Z.H. and M.L.
supervised the research, including method-ology development, as well as manuscript structure,
writing and re-vision. M.L. drafted the manuscript. Z.H., F.S., M.G., P.X. and Y.Y. prepared the
data and wrote this paper. All authors have read and agreed to the published version of the
manuscript.
**Funding:** This work was financially supported by the Second Tibetan Plateau Scientific
Expedi-tion and Research (STEP) program (Grant No. 2019QZKK0103), the National Natural
Science Foundation of China (Grant No. 41675106) and Scientific Research Project of Chengdu
University of Information Technology (KYTZ201721).
**Data Availability Statement:** The surface energy fluxes data used in this article are available
online at https://data.tpdc.ac.cn/en/data/b9ab35b2-81fb-4330-925f-4d9860ac47c3/ (Ma, 2020.
accessed on 18 October 2020). The radiosonde data used in this article are available online at
https://data.tpdc.ac.cn/zh-hans/data/70edaec5-8418-44cd-afc4-fb089f7bf413/ (Li, 2022. accessed
on 18 Mar 2022). The variables used in the reanalysis of the ERA-5 data
(https://www.ecmwf.int/en/about/media-centre/science-blog/2017/era5-new-reanalysis-weather-
and-climate-data, accessed on 18 October 2020).
**Acknowledgments**: This work was financially supported by the Second Tibetan Plateau
Scientific Expedition and Research (STEP) program (Grant No. 2019QZKK0103), the National
Natural Sci-ence Foundation of China (Grant No. 41675106, 41805009), National key research
and develop-ment program of China (2017YFC1505702) and Scientific Research Project of
Chengdu University of Information Technology (KYTZ201721).





**Conflicts of Interest:** The authors declare no conflict of interest.

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




Table 1 observational station information

| Station | Latitude | Longitude | Altitude | Type of underlying surface |
|---------|----------|-----------|----------|----------------------------|
| Mount Everest | 28.21°N | 86.56°E | 4276 m | Mainly bare land, with sparse and dwarf vegetation |
| Nyingchi | 29.77°N | 94.74°E | 3326 m | Alpine meadow |
| Namco | 30.77°N | 90.96°E | 4730 m | Alpine meadow |
| Nagqu | 31.37°N | 91.90°E | 4509 m | Alpine meadow |
| Shiquan River | 32.49°N | 80.10°E | 4278 m | Flatter bare soil |







Table 2 The surface heating field at the height of the boundary layer at each station in 2014

(the observation values for Nyingchi in June and August are missing "-", and some of the
observation values for Nam Co in November are missing "-")

| Station | Month | Convective boundary layer height (m) | Observed sensible heat flux (W·m⁻²) | Observed latent heat flux (W·m⁻²) | ERA5 sensible heat flux (W·m⁻²) | ERA5 latent heat flux (W·m⁻²) |
|---|---|---|---|---|---|---|
| Mount Everest | June | 3000 | 282.56 | 70.47 | 177.73 | 242.19 |
| | August | 1700 | 181.37 | 115.00 | 140.15 | 282.18 |
| | November | 4500 | 237.56 | 11.54 | 170.17 | 62.65 |
| Nyingchi | June | 2100 | - | - | 144.51 | 237.70 |
| | August | 1000 | - | - | 75.22 | 151.64 |
| | November | 3000 | 207.11 | 109.69 | 133.37 | 66.07 |
| Nam Co | June | 2200 | 200.21 | 70.38 | 220.64 | 279.35 |
| | August | 950 | 137.79 | 259.06 | 112.67 | 169.37 |
| | November | 2400 | - | - | 107.37 | 94.94 |





Table 3 Surface heating field and the height of the boundary layer at each station in 2019

| Station | Month | Convective boundary layer height (m) | ERA5 sensible heat flux (W·m⁻²) | ERA5 latent heat flux (W·m⁻²) |
|---|---|---|---|---|
| Mount Everest | May | 2800 | 244.88 | 104.89 |
| | July | 2000 | 118.59 | 276.25 |
| | October | 3250 | 142.07 | 77.70 |
| Nyingchi | May | 2250 | 98.34 | 91.34 |
| | July | 2100 | 154.84 | 287.51 |
| | October | 700 | 84.95 | 65.36 |
| Nagqu | May | 2760 | 205.77 | 220.02 |
| | July | 1650 | 165.98 | 374.23 |
| | October | 2700 | 61.14 | 114.59 |
| Shiquan River | May | 2650 | 284.54 | 43.86 |
| | July | 2000 | 118.26 | 261.07 |
| | October | 3500 | 215.09 | 7.74 |





**Figure captions**
Fig. 1 the locations of the sounding stations
Fig. 2 Potential temperature profiles in the convective boundary layer observed at the (a)
Mount Everest, (b) Nyingchi and (c) Nam Co stations in June, August, and November 2014 and
at the (d) Shiquan River station in May, July, and October 2019
Fig. 3 Potential temperature profiles for the Mount Everest, Nyingchi, and Nam Co stations
in June, August, and November 2014 and for the Shiquan River station in May, July, and October

2019

Fig. 4 Specific humidity profiles for the Mount Everest (a), Nyingchi (b), and Nam Co (c)
stations in June, August, and November 2014 and for the Shiquan River (d) station in May, July,
and October 2019
Fig. 5 Wind speed profiles for the Mount Everest (a), Nyingchi (b), and Nam Co (c) stations
in June, August, and November 2014 and for the Shiquan River (d) station in May, July, and
October 2019
Fig. 6 Wind direction profiles for the Mount Everest, Nyingchi, and Nam Co stations in
June, August, and November 2014 and for the Shiquan River station in May, July, and October

2019

Fig. 7 Comparison of the surface heat flux between the reanalysis data and observations on
the TP (unit: W·m$^{-2}$)
Fig. 8 Compare scatter plots of the surface heat flux between the reanalysis data and
observations at the Mount Everest station (a–b), the Nyingchi station (c–d) and the Nam Co
station (e–f) (unit: W·m$^{-2}$).
Fig. 9 Latent and sensible heat fluxes at the Mount Everest, Nyingchi, Nagqu, and Shiquan
River stations in 2019 based on the ERA5 reanalysis data (unit: W·m$^{-2}$)
Fig. 10 Vertical velocities at the (a–c) Mount Everest, (d–f) Nyingchi, and (g–i) Namco
stations in 2014 and at the (j–l) Shiquan River station in 2019 based on ERA5 reanalysis data
(Unit: Pa s$^{-1}$)



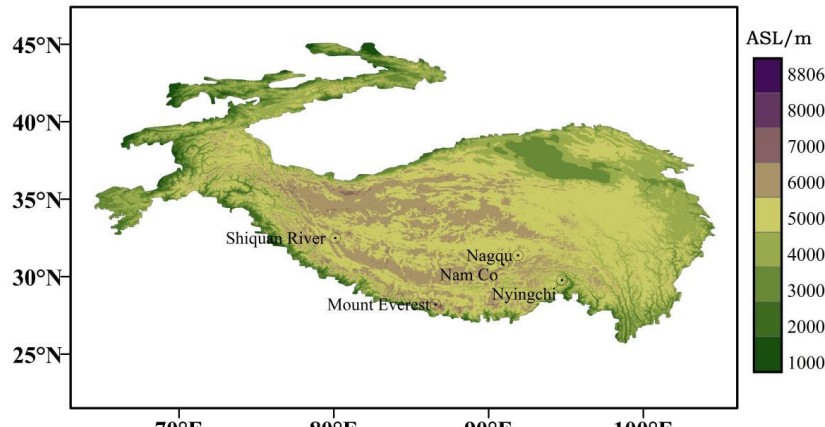


Figure 1 Location distribution of sounding stations






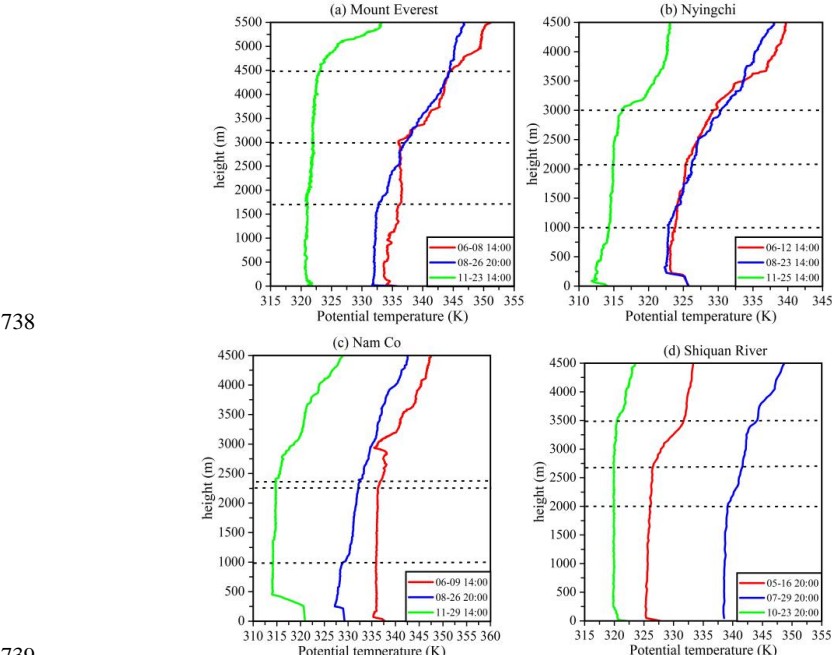



Fig.2 Potential temperature profiles of the convective boundary layer observed at the three
stations of Mount Everest(a), Nyingchi(b) and Nam Co(c) in June, August and November2014,
and for the Shiquan River(d) station in May, July, and October 2019



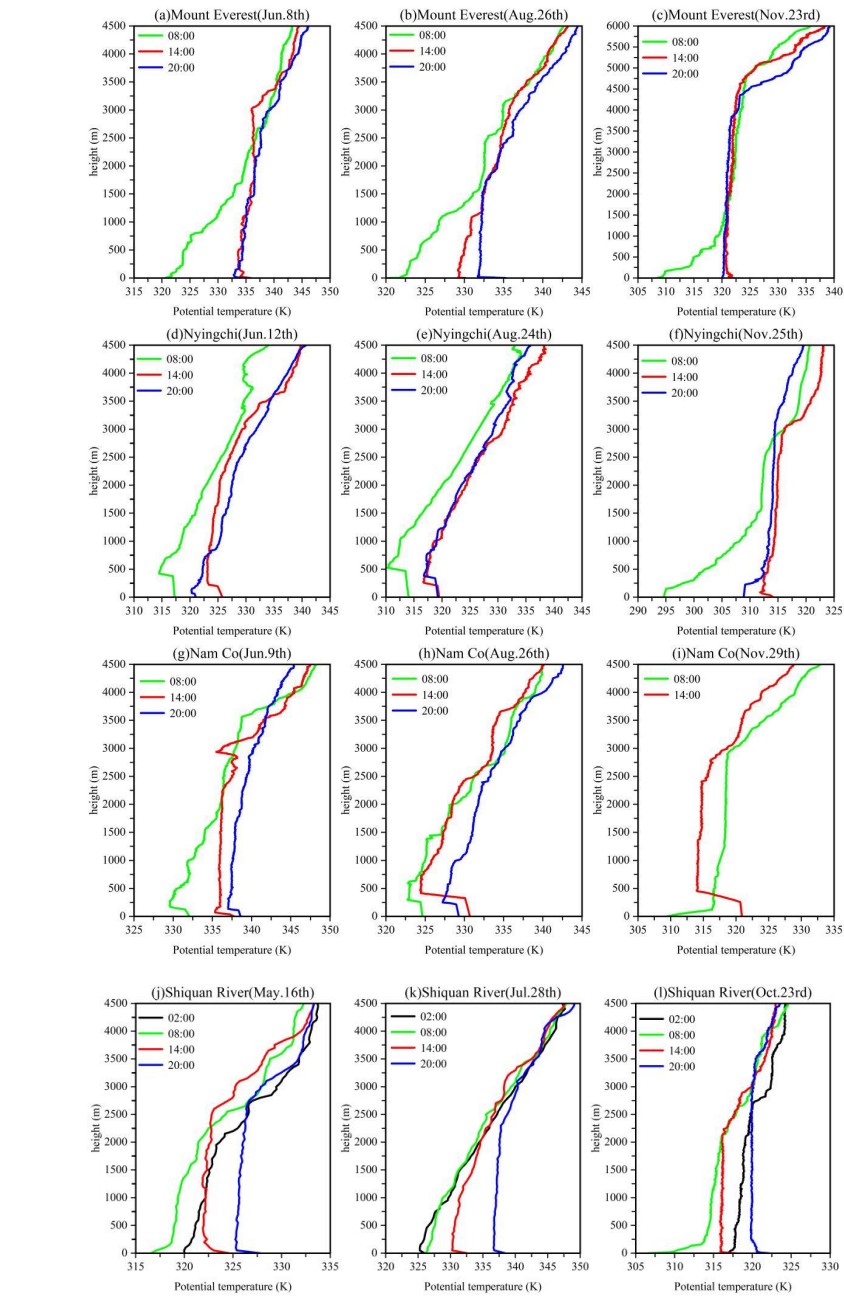



Fig.3 Potential temperature profiles of Mount Everest, Nyingchi and Nam Co in June,

August and November 2014, and for the Shiquan River station in May, July and October 2019




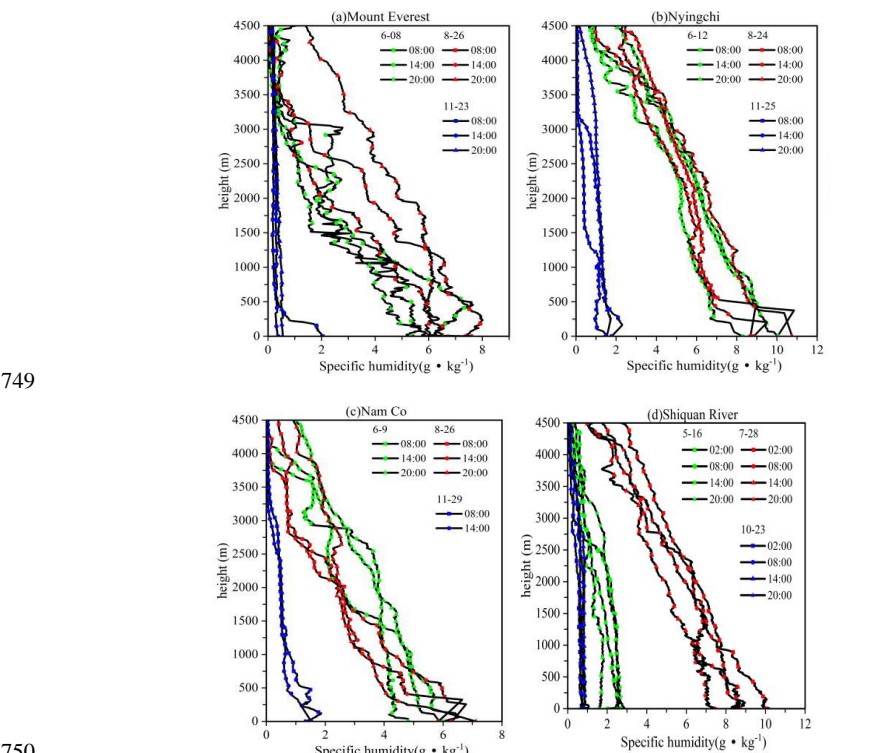



Fig.4 Specific humidity profiles for the Mount Everest (a), Nyingchi (b), and Nam Co (c)
stations in June, August, and November 2014 and for the Shiquan River (d) station in May, July,
and October 2019



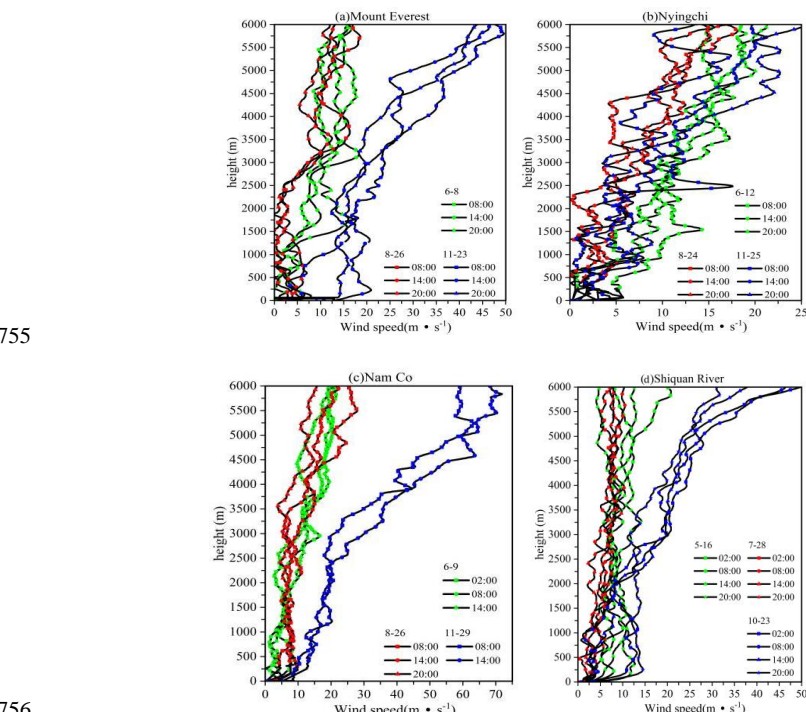

Fig. 5 Wind speed profiles for the Mount Everest (a), Nyingchi (b), and Nam Co (c) stations

in June, August, and November 2014 and for the Shiquan River (d) station in May, July, and

October 2019



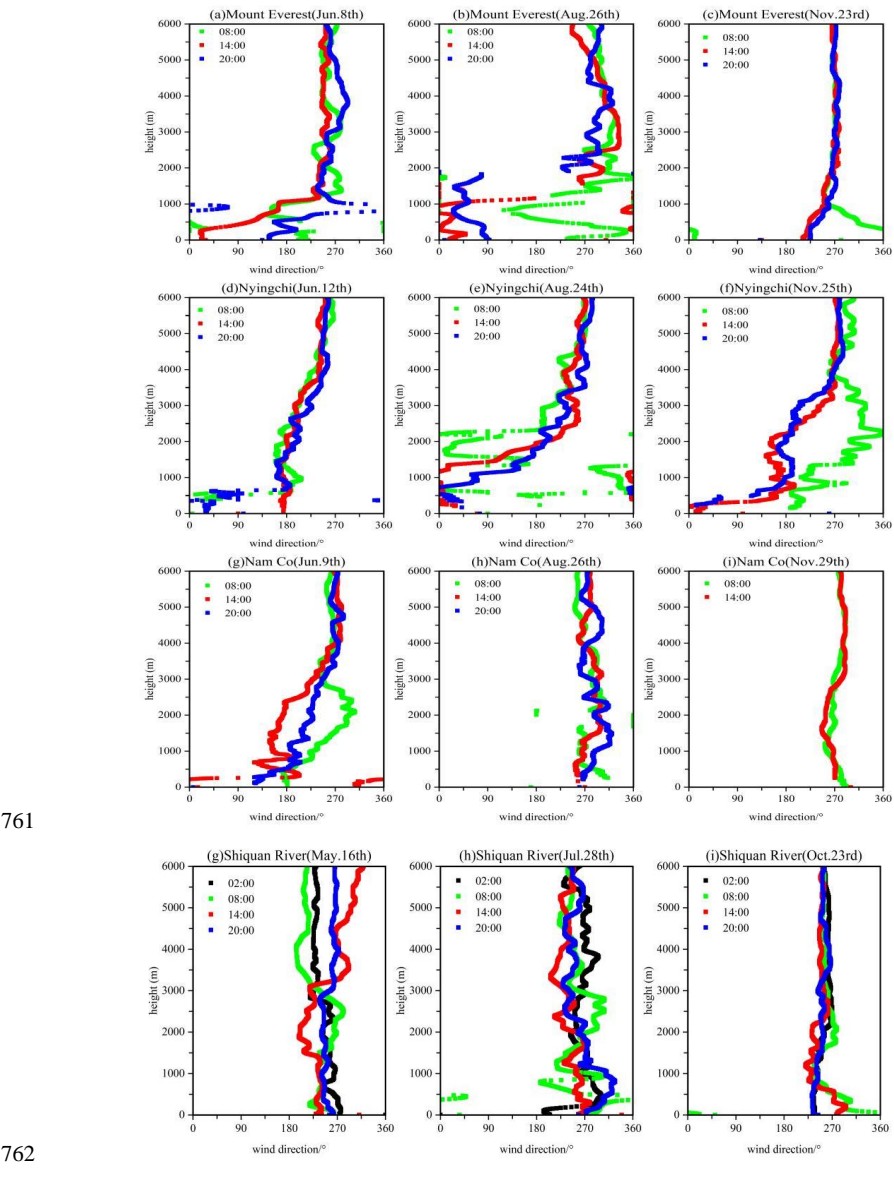



Fig. 6 Wind direction profiles of Mount Everest, Nyingchi and Nam Co in June, August and

November 2014, and Shiquan River in May, July and October 2019




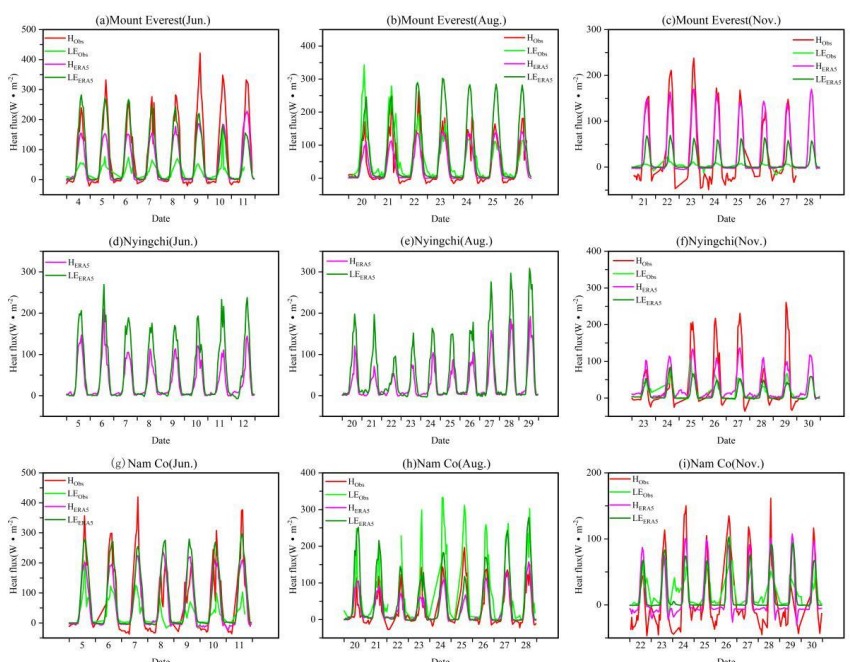


Fig.7 Comparison of the surface heat fluxes between the reanalysis data and observation on
the TP (unit: W·m⁻²)

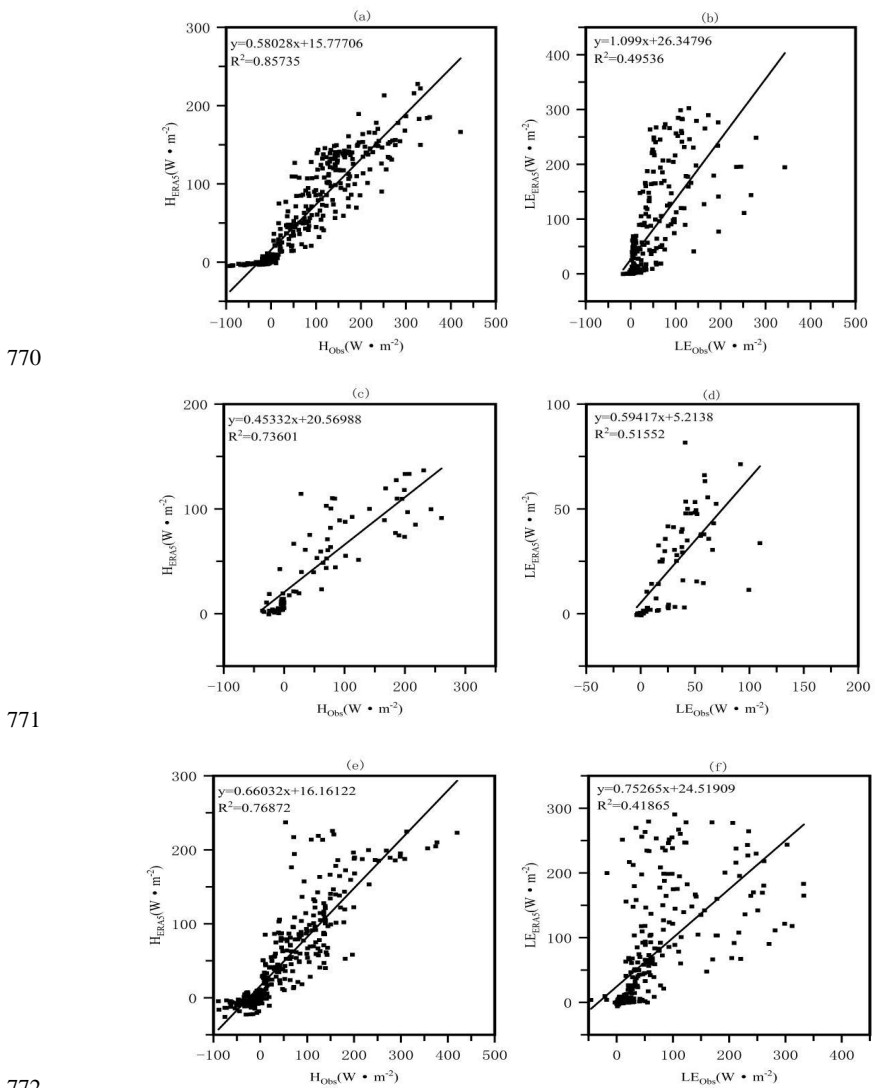




Fig.8 Compare scatter plots of the surface heat flux between the reanalysis data and

observations at the Mount Everest station (a–b), the Nyingchi station (c–d) and the Nam Co

station (e–f) (unit: W·m$^{-2}$).






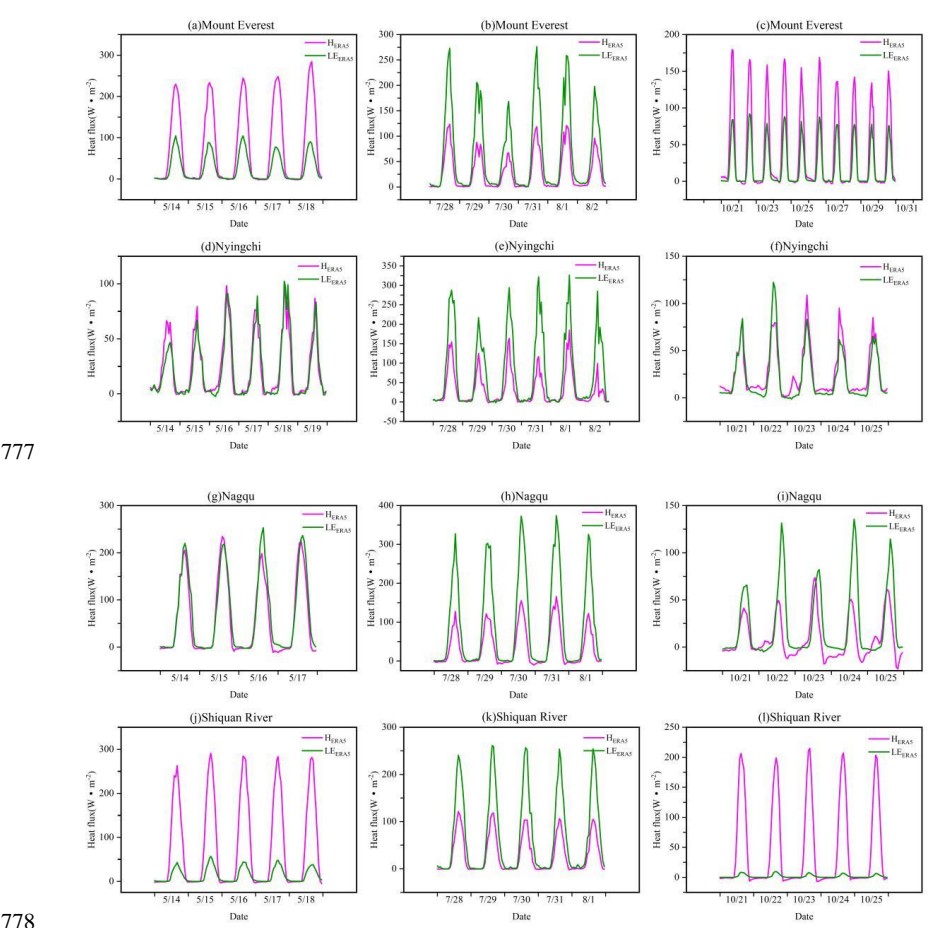



Fig.9 the latent heat and the sensible heat flux of Mount Everest, Nyingchi, Nagqu, and

Shiquan River based on the reanalysis data of ERA5 in 2019 (unit:W·m⁻²)


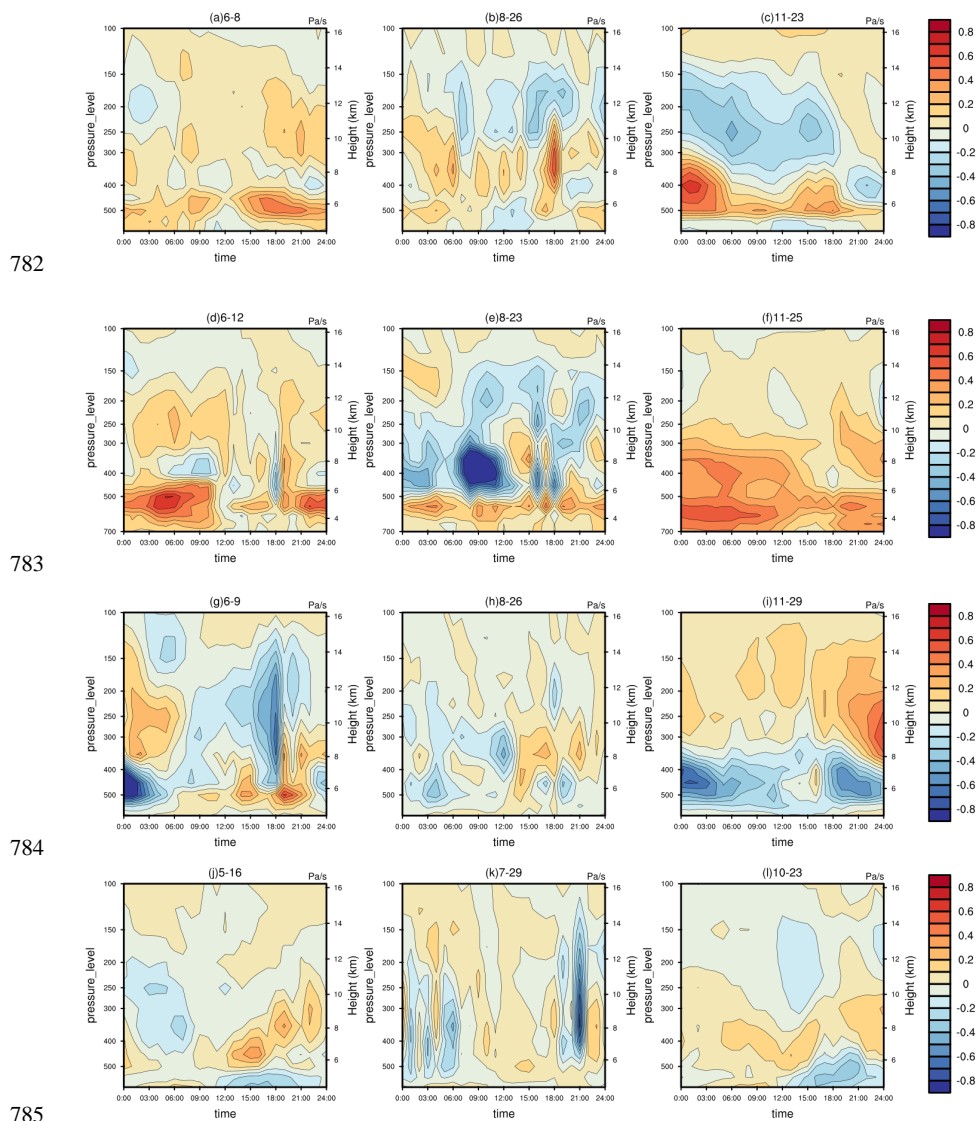





Fig. 10 Vertical velocities of Mount Everest(a~c), Nyingchi(d~f) , Namco(g~i) in 2014 and

Shiquan River(j~l) in 2019 based on ERA5 reanalysis data (Unit: Pa s$^{-1}$)
