# Peer review of "Measurement report: Structure of the atmospheric boundary layer 1 and its relationship with the land-atmosphere interaction on the 2 Tibetan Plateau 3 4 5 Maoshan Li11, Wei Fu2, Na Chang1, Ming Gong1, Pei Xu1, Yaoming Ma 3, Z"

_Atmospheric Chemistry and Physics, 2022_

## Author Comment (AC2)

We are thankful to the reviewers for their valuable comments on the paper. Below we provide the responses to the comments and questions raised by point to point. Modifications and improvements are incorporated in the revised manuscript as mentioned below for each of the comments. For easy visualization, the responses to the reviewers' comments in bold are provided below

**The authors studied the boundary layer structure in the Tibetan Plateau on the basis of sounding measurements in 2014 and 2019. However, there is nothing new in this manuscript, and only a few sounding profiles were presented. Most literature cited was published twenty years ago, the authors failed to clarify the motivation for the work presented. The authors tried to understand the different influence of the southern branch of the westerly wind and Asian monsoon, while the large-scale synoptic conditions were not presented and analyzed. Also, the authors mentioned the local terrain wind systems (e.g., westerly jet, glacial wind) in the Result section, while no observational evidences were presented.**

**Response,** Thank you very much for your valuable comments

This article is just a measurement report. Its' purpose is to introduce the different characteristics of the atmospheric boundary layer structure under the coordinated action of the westerly and monsoons on the Tibetan Plateau. The experimental design selects 2014 and 2019 accordingly to carry out the observation experiment in the intensive period and analyzes the effect of the ground heating field on the atmospheric boundary combined with the ground flux observation. The influence of the structure of the atmospheric boundary layer in the westerly and monsoon periods can be revealed.

Previous studies have shown that there is glacial wind in the Mount Everest region (Zou, et al., 2007; Sun, et al., 2017; Sun, et al., 2018). In the future, model simulations will be used to further study the impact of the local circulation field on the structure of the atmospheric boundary layer. Similarly, the impact of high-altitude westerly jets can also be studied through model simulations, thereby revealing the changing mechanism of the atmospheric boundary layer structure.

Zou H., Zhou L., Ma S., Li P., Li A., Huo C.: Observation of Diurnal Circulation on the Northern Slope of Mt. Qomolangma,Plateau Meteorology, 26(6):1123-1140,2007

Sun F.,Ma Y.,Hu Z., et al. Observation of Strong Winds on the Northern Slopes of Mount Everest in Monsoon Season[J]. Arctic Antarctic & Alpine Research, 49(4):687-697, doi:10.1657/AAAR0016-010, 2017.

Sun, F., Yao M., et al. Mechanism of Daytime Strong Winds on the Northern Slopes of Himalayas, near Mount Everest: Observation and Simulation[J]. Journal of Applied Meteorology and Climatology, 57(2):255-272, 2018.

We have added nearly 20 years of relevant research as follows, and references have been added.

Che, J. and Zhao, P.: Characteristics of the summer atmospheric boundary layer height over the Tibetan Plateau and influential factors, Atmospheric Chemistry and Physics, 21, 5253-5268, 10.5194/acp-21-5253-2021, 2021.

Che, J. and Zhao, P.: Characteristics of the summer atmospheric boundary layer height over the Tibetan Plateau and influential factors, Atmospheric Chemistry and Physics, 21, 5253-5268, 10.5194/acp-21-5253-2021, 2021.

Chen, X. L., Juan, A. Añel., Su, Z. B., Laura, de. La. Torre., Hen□nie, Kelder., Jacob, van. Peet., and Ma, Y. M.: The deep atmo□spheric boundary layer and its significance to the stratosphereand troposphere exchange over the Tibetan Plateau, PLoS. ONE, 8, e56909, https://doi.org/10.1371/journal.pone.0056909, 2013.

Chen, X. L., Škerlak, B., Rotach, M. W., Juan, A. Anel., Su, Z. B., Ma, Y. M., and Li, M. S.: Reasons for the ex□tremely high-ranging planetary boundary layer over the West□ern Tibetan Plateau in winter, J. Atmos. Sci., 73, 2021–2038, https://doi.org/10.1175/jas-d-15-0148.1, 2016

Davy, R.: The Climatology of the Atmospheric Boundary Layer in Contemporary Global Climate Models, Journal of Climate, 31, 9151-9173, 10.1175/jcli-d-17-0498.1, 2018.

Davy, R. and Esau, I.: Differences in the efficacy of climate forcings explained by variations in atmospheric boundary layer depth, Nat Commun, 7, 11690, 10.1038/ncomms11690, 2016.

Dirmeyer, P. A., Wang, Z., Mbuh, M. J., and Norton, H. E.: Intensified land surface control on boundary layer growth in a changing climate, Geophysical Research Letters, 41, 1290-1294, 10.1002/2013gl058826, 2014.

Guo, J., Miao, Y., Zhang, Y., Liu, H., Li, Z., Zhang, W., He, J., Lou, M., Yan, Y., Bian, L., and Zhai, P.: The climatology of planetary boundary layer height in China derived from radiosonde and reanalysis data, Atmospheric Chemistry and Physics, 16, 13309-13319, 10.5194/acp-16-13309-2016, 2016.

Li, N., Zhao, P., Wang, J., and Deng, Y.: The Long-Term Change of Latent Heat Flux over the Western Tibetan Plateau, Atmosphere, 11, 10.3390/atmos11030262, 2020.

Shi, Y., Wei, J., Qiao, Z., Zhao, J., and Wang, G.: Atmospheric Exploration of the Qinghai–Tibet Plateau during the East Asian Winter Monsoon (EAWM) from a Ground-Based Microwave Radiometer, Atmosphere, 13, 10.3390/atmos13040549, 2022.

Zhang, K., Wu, Y., Wang, F., Li, X., Cui, S., Zhang, Z., Wu, X., Weng, N., Luo, T., and Huang, Y.: Characteristics of the vertical structure of the atmospheric turbulence in the Tibetan Plateau, Science China Earth Sciences, 65, 1368-1378, 10.1007/s11430-021-9922-0, 2022.

Zhao, P., Zhou, X., Chen, J., Liu, G., and Nan, S.: Global climate effects of summer Tibetan Plateau, Science Bulletin, 64, 1-3, 10.1016/j.scib.2018.11.019, 2019.

Wang, Y., Xu, X., Liu, H., Li, Y., Li, Y., Hu, Z., Gao, X., Ma, Y., Sun, J., Lenschow, D. H., Zhong, S., Zhou, M., Bian, X., and Zhao, P.: Analysis of land surface parameters and turbulence characteristics over the Tibetan Plateau and surrounding region, Journal of Geophysical Research: Atmospheres, 121, 9540-9560, 10.1002/2016jd025401, 2016.

Xiang, J., Zhou, J., and Huang, S.: The boundary layer height obtained by the spline numerical differentiation method using COSMIC GPS radio occultation data: A case study of the Qinghai-Tibet Plateau, Journal of Atmospheric and Solar-Terrestrial Physics, 215, 10.1016/j.jastp.2020.105535, 2021.

---

## Author Comment (AC3)

**RC1**

We are thankful to the reviewers for their valuable comments on the paper. Below we provide the responses to the comments and questions raised by point to point. Modifications and improvements are incorporated in the revised manuscript as mentioned below for each of the comments. For easy visualization, the responses to the reviewers' comments in bold are provided below

**This study used the sounding, surface heat flux and reanalysis data to characterize the PBL structures of 5 stations in Tibetan Plateau (TP) under differentwind flow conditions (south branch of westerly wind VS summer monsoon).**

**The major weakness of this study is a lack of physical explanation of the PBL structures from the data analysis. The sounding data analysis presents only a few days to characterize the PBL structure under the influence of monsoon flow, plateau flow, and westerly flow. The results from the limited data analysis are not sufficient. A seasonal and long-term data analysis is required to enhance the validity of this study.**

The review comments for our article "Measurement report: Structure of the atmospheric boundary layer and its relationship with the land-atmosphere interaction on the Tibetan Plateau" have been received. First, we would like to thank the reviewers for their valuable revisions. At the same time, we would like to thank the editor for their hard work in improving this article. We carefully read the review comments and adopted and responded to the comments of the reviewers. Now we will explain the response results one by one in combination with the review comments:

1. **P3, line 82-84, It is found that the thermal structure of the
Plateau Atmospheric boundary layer is abnormal, the development of the convective boundary
layer is deep, and the dynamic mechanism of Ekman "suction pump" in the plateau boundary layer.
What do you mean "abnormal"? Please explain "suction pump".**

   **Response,** The thermal anomaly is based on the change of the surface temperature change index to determine the warm and cold anomalous years. The mechanism by which the effects of boundary layer momentum fluxes are communicated directly to the neighboring (essentially inviscid) fluid. The Tibetan Plateau is in central Asia. It is the highest plateau with the most complex terrain in the world. The average altitude of the Tibetan plateau is above 4000m, which is about one-third of the troposphere. A thermal forcing disturbance source towering into the free atmosphere is formed through the transport of radiation, sensible and latent heat in the near-surface and boundary layers. The energy and material transport between earth and gas is an important driving force for the development of the atmospheric boundary layer, while the dynamic and thermal effects of the Qinghai-Tibet Plateau on the atmosphere mainly affect the free atmosphere gradually through the near-surface layer and boundary layer of the plateau. The intense solar radiation in the plateau region provides enough heating to the surface, causing the bottom of the atmospheric boundary layer to

receive strong ground heating, resulting in abnormal thermal structure of the atmospheric boundary layer in the plateau.

**Ekman "suction pump"**, Due to the existence of geostrophic wind vorticity in the free atmosphere, there is an upward movement in the atmospheric boundary layer. Through this mechanism, momentum、heat and water vapor are exchanged between the atmospheric boundary layer and the free atmosphere.

2. **P3, line 85, why there are extreme values?**

**Response,** The author uses the detection data of the tethered airship to analyze the multi-extreme phenomenon of wind in the boundary layer. The possible reason is that the atmosphere is unstable due to the heating of the ground in the lower layer, and the temperature inversion layer in the upper layer blocks the downward transmission of momentum (Li et al., 2000).

Li, J., Hong, Z., Sun, S. : An Observational Experiment on the Atmospheric Boundary Layer

in Gerze Area of the Tibet Plateau[J]. Chinese Journal of Atmospheric Sciences, (03):301-312, 2000.

3. **P4, "strengthening period", unclear description.**

**Response,** It should be 'Intensive Observation Period (IOP)', it was revised in the revised manuscript.

4. **Line 94, why PBL height is high in west and low in east?**

**Response,** The spatial distribution of the boundary layer height on the Tibetan Plateau is jointly determined by the dynamic and thermal characteristics of the plateau. The authors used the COSMIC data from 2007 to 2013 to determine the boundary layer height by calculating the minimum gradient of atmospheric refractive index, and checked the results with radiosonde data. (GPS occultation data profiles have the characteristics of high precision, high vertical resolution, and global coverage. Due to the long wavelength, they can penetrate clouds and rain, and can still be observed under weather conditions. GPS occultation radio signals can penetrate the atmosphere and reach lower altitudes, and because of the high data accuracy, boundary layer information can be accurately described. GPS occultation data is beneficial to the overall understanding of the distribution characteristics of the plateau boundary layer.) The height of the upper boundary layer on the Tibetan Plateau is high in the west and low in the east. The height of the western boundary layer is mainly 1.8-2.3 km, while that of the eastern boundary layer is 1.4-1.8 km. The center of the maximum value is in the southwest of the plateau. The spatial distribution of boundary layer height detected by COSMIC is basically consistent with that of ERA-Int, which is high in the west and low in the east.

5. **What is the similarities and dissimilarities of this study comparing with the previous studies? What is the importance and unique results from this study?**

   **Response,** Due to the vast and complex terrain of the plateau, previous studies were mostly based on satellite data or reanalysis data to analyze the height of the overall boundary layer of the plateau, or radiosondes focused on a specific site. This paper summarizes and analyzes the characteristics of boundary layer height, humidity, wind direction and wind speed at different azimuth stations of the plateau and summarizes the different regional characteristics of the plateau boundary layer. According to the characteristic analysis of ground sensible heat, latent heat flux and vertical velocity field, the relationship between boundary layer structure and sensible heat, latent heat flux and vertical velocity field is studied. To gain an in-depth understanding of the variation characteristics of the plateau atmospheric boundary layer height, specific humidity, wind direction and wind speed in the coordination area between the south branch of the westerly wind and the plateau monsoon, it will provide a certain theoretical basis for the forecast and early warning of the plateau weather and climate under the background of climate change in the future.

6. **Line 142, is PBL height determined by subjective personal inspection?**

   **Response,** Regarding the methods for determining the height of the boundary layer. At present, there are many methods commonly used, such as the potential temperature gradient method, the Holzworth method (dry adiabatic method, the gas block method), the Richardson method, and the potential temperature profile method. The data in this paper have tried to use two calculation methods, the potential temperature gradient method and the Richardson method, but the results are not satisfactory. Therefore, the determination of the boundary layer height in this paper is mainly based on the potential temperature profile method, and the height where the gradient is obviously discontinuous is regarded as the height of the atmospheric boundary layer. During the day, the height of the roof with strong inversion is taken as the height of the convective boundary layer. At night, the height of the ground inversion layer is taken as the height of the stable boundary layer. From the perspective of dynamic action, the height of the stable boundary layer at night can be regarded as the height of the stable boundary layer according to the height of the maximum wind speed. At the same time, from the perspective of material distribution, the height of the convective boundary layer can be regarded as the height of the convective boundary layer according to the height where the specific humidity decreases rapidly. We will determined ABLH using Che's method (Che et al., 2019).

   Che, J. and Zhao, P.: Characteristics of the summer atmospheric boundary layer height over the Tibetan Plateau and influential factors, Atmospheric Chemistry and Physics, 21, 5253-5268, 10.5194/acp-21-5253-2021, 2021.

7. **Line 157, this study only utilized limited observation data in 2014 and 2019.**

   **Response,** The purpose of this paper is to introduce the different characteristics of the atmospheric boundary layer structure under the coordinated action of the westerly and monsoons on the Qinghai-Tibet Plateau. The experimental design selects 2014 and 2019 accordingly to carry out the observation experiment in the enhanced period and analyzes the effect of the ground heating field

on the atmospheric boundary combined with the ground flux observation. The influence of the structure of the atmospheric boundary layer in the westerly and monsoon periods can be revealed.

8. **This study only utilized limited sounding observations to characterize the PBL structures. I suggest you expanding the analysis by including the regular sounding observations (twice a day 00 and 12 UTC) for longer period (3 to 5 years) to enhance the representativeness of the results.**

   **Response,** Thank you very much for your valuable comments. In the later period, we will strive to obtain longer-term sounding data to describe the evolution of the plateau atmospheric boundary layer structure in detail. In addition, due to some objective reasons, conventional observational data cannot be obtained. In view of the observational purpose of our experiment, only two-year intense observational periods data were selected.

9. **Discussions in session 3 and 4 are very descriptive and lacking in any in-depth interpretation. It's very difficult to understand the whole discussions. The physics responsible for characterizing the structures are not discussed.**

   **Response,** This article is just a measurement report. Its purpose is to introduce the different characteristics of the atmospheric boundary layer structure under the coordinated action of the westerly and monsoons on the Tibetan Plateau. The experimental design selects 2014 and 2019 accordingly to carry out the observation experiment in the intensive period and analyzes the effect of the ground heating field on the atmospheric boundary combined with the ground flux observation. The influence of the structure of the atmospheric boundary layer in the westerly and monsoon periods can be revealed.

10. **Line 168, why the low-level interval of radiosonde data is large?**

   **Response,** Thank you very much for your valuable comments. The original sounding data has been reprocessed, and the corresponding potential temperature and specific humidity maps have been modified and analyzed as follow and in the revised version.

[Figure]

Fig.3 Potential temperature profiles of Mount Everest, Nyingchi and Nam Co in June, August and November 2014, and for the Shiquan River station in May, July and October 2019

**11. Line 194-196, how the westerly jet affect inversion stratification?**

**Response,** The occurrence of the southwesterly wind during non-monsoon was in good consistency with high values of westerly wind at high levels over this region and confirmed to be driven by the strong westerly jet aloft at Everest Mount. Station (Sun, et al., 2018). We will explore the mechanism of the westerly jet affect inversion stratification using WRF model in the near future.

Sun, F., Yao M., et al. Mechanism of Daytime Strong Winds on the Northern Slopes of Himalayas, near Mount Everest: Observation and Simulation[J]. Journal of Applied Meteorology and Climatology, 57(2):255-272, 2018.

**12. Line 199, what is super-insulation layer?**

**Response,** In the atmosphere, there is a super-insulation layer in the vertical temperature change, that is, there is a vertical temperature gradient, and the temperature increases by more than 1 °C for every 100 meters of height. This phenomenon is not an accidental existence, but generally exists in the atmosphere.

**13. Line 214, "Stable Boundary Layer" height reaches 1500 m? Isn't 1500 m too high for a SBL?**

**Response,** The original sounding data has been reprocessed, and the corresponding potential temperature and specific humidity maps have been modified and analyzed as follow and in the revised version. According to the analysis of observational data, the height of the stable boundary layer reaches only 500 m.

**14. Line 243, why convective BL height reached highest point at hour 20:00?**

**Response,** The height of the convective boundary layer in the Shiquan River area often reaches the highest point at 20:00. According to the comprehensive observation results four times a day, the height of the convective boundary layer in the Shiquan River area is the highest at 20:00.

**15. Line 284, how did you tell there is formation of valley wind?**

**Response,** Previous studies have shown that there is glacial wind in the Mount Everest region (Zou, et al., 2007; Sun, et al., 2017; Sun, et al., 2018). In the future, model simulations will be used to further study the impact of the local circulation field on the structure of the atmospheric boundary layer. Similarly, the impact of high-altitude westerly jets can also be studied through model simulations, thereby revealing the changing mechanism of the atmospheric boundary layer structure.

Zou H., Zhou L., Ma S., Li P., Li A., Huo C.: Observation of Diurnal Circulation on the Northern Slope of Mt. Qomolangma,Plateau Meteorology, 26(6):1123-1140,2007

Sun F.,Ma Y.,Hu Z. , et al. Observation of Strong Winds on the Northern Slopes of Mount Everest in Monsoon Season[J]. Arctic Antarctic & Alpine Research, 49(4):687-697, doi:10.1657/AAAR0016-010, 2017.

Sun, F., Yao M., et al. Mechanism of Daytime Strong Winds on the Northern Slopes of Himalayas, near Mount Everest: Observation and Simulation[J]. Journal of Applied Meteorology and Climatology, 57(2):255-272, 2018.

**16. Line 305, how the glacial wind is formed? How to tell there is formation of glacial wind?**

**Response,** Glacier breeze means that in the glacier valley, the relatively stable and sinking cooling air flow on the glacier surface moves along the ice towards the front of the glacier, forcing the warmer air in the edge of the ice to rise and produce convection exchange, forming a movement from the glacier surface to the ice. The wind blowing from the edge. If the glacier is large enough, the glacier wind can prevail throughout the day, and it can also extend farther from the front of the glacier, and the glacier wind thickness is also large. The temperature difference between the air temperature on the glacier surface and the air temperature at the same height in the valley varies daily. Although the wind direction of the glacier wind remains unchanged throughout the day, the wind speed varies daily with a 24-hour cycle.

Previous studies have shown that there is glacial wind in the Mount Everest region (Zou, et al., 2007; Sun, et al., 2017; Sun, et al., 2018). In the future, model simulations will be used to further study the impact of the local circulation field on the structure of the atmospheric boundary layer. Similarly, the impact of high-altitude westerly jets can also be studied through model simulations, thereby revealing the changing mechanism of the atmospheric boundary layer structure.

Zou H., Zhou L., Ma S., Li P., Li A., Huo C.: Observation of Diurnal Circulation on the Northern Slope of Mt. Qomolangma,Plateau Meteorology, 26(6):1123-1140,2007

Sun F. , Ma Y. , Hu Z. , et al. Observation of Strong Winds on the Northern Slopes of Mount Everest in Monsoon Season[J]. Arctic Antarctic & Alpine Research, 49(4):687-697, doi:10.1657/AAAR0016-010, 2017.

Sun, F., Yao M., et al. Mechanism of Daytime Strong Winds on the Northern Slopes of Himalayas, near Mount Everest: Observation and Simulation[J]. Journal of Applied Meteorology and Climatology, 57(2):255-272, 2018.

**17. Line 311-314, unclear description.**

**Response,** they were revised 'In summary, at all the stations, the wind direction and wind speed in the lower level were greatly affected by the topography and geographical location, while the changes in the wind direction and wind speed in the upper level were related to the large-scale westerly circumfluence and the Asian monsoon.'

**18. Line 346-349, there is apparent bias between observation data and ERA5 data.**

**Response,** There is indeed bias between the observed data and the ERA5 data. The trend of the observed data and the ERA5 data is consistent, showing a unimodal change, which has certain indicative significance.

**19. Line 331, "under coordinated action of the westerly wind and monsoon", unclear description.**

**Response,** it was revised 'During different periods of westerly and monsoon';

**20. The structure of the presentation needs to be better organized.**

**Response,** thank you for your comment. It was re-organized in the revised manuscript.

**21. The writing needs to be improved.**

**Response,** LetPub edited and revised the English writing (the certificate can be seen as follow).

[Figure]

**Certificate of English Language Editing**

**Manuscript Title:**
Structure of the atmospheric boundary layer and its relationship with the land-atmosphere interaction on the Tibetan Plateau

**Date of Revision:**
March 23, 2022

Abstract:
There is a deep atmospheric boundary layer on the Tibetan Plateau (TP) that has always been of interest to researchers. The variation in the atmospheric boundary layer under the influence of the southern branch of the westerly wind and that of the Asian monsoon was analyzed using sounding data collected in 2014 and 2019. Then, the hourly high-resolution comprehensive observation data for the land-atmosphere interaction on the TP and the ERA5 reanalysis data were used to study the influence of the atmospheric boundary layer's structure in Mount Everest, Nyingchi, Nam Co, Nagqu, and Shiquan River regions. The results show that the height of the convective boundary layer observed at the Mount Everest, Nyingchi, Nam Co, Nagqu, and Shiquan River stations on the TP under the influence of the southern branch of the westerly wind was higher than that during the Asian monsoon season. The height of the convective boundary layer in the Shiquan River area was often highest at 20:00. The structure of the boundary layer in the Mount Everest area was often affected by the westerly jets and glacial winds. The inversion layer ...

This document certifies that the manuscript listed above was copy edited for English language by LetPub, with regard to grammar, punctuation, spelling, and clarity. All of our language editors are native English speakers with long-term experience in editing scientific and technical manuscripts. We are committed to leveling the playing field for researchers whose native language is not English.

- Documents receiving this certification should be regarded as having undergone professional editorial revision for English language before submission. However, the authors may accept or reject LetPub's suggestions and changes at their own discretion and LetPub does not have editorial control over the submitted documents.
- The language quality of the submitted document is the sole responsibility of the submitting authors subject to those authors' adherence to LetPub's revisions and instruction. LetPub's provision of service does not constitute a guarantee or endorsement of the authors' work herein.
- Neither the research content nor the authors' intended meaning were altered in any way during the editing process.
- If you have any questions or concerns about this edited document, please contact us at support@letpub.com

[Figure]

LetPub is an author service brand owned and operated by Accdon LLC. Headquartered in the Boston area, we are a full-spectrum author services company with a large team of US-based certified language and scientific editors, ISO 17001 accredited translators, and professional scientific illustrators and animators. We advocate ethical publication practices and are an official member of the Committee on Publication Ethics (COPE).

For more information about our company, services, and partnership programs, please visit www.letpub.com.

**22. Please remove the general descriptions in the manuscript to enhance the understanding of the importance from this study.**

**Response,** thank you for your good suggestion. We will revise the manuscript according to your comments.

23. **Sessions 3 and 4 and conclusions of the current form are lengthy and tedious descriptions and must be rewritten to enhance the interest of the reader.**

    **Response,** thank you for your good suggestion. We revised the manuscript according to your comments.

24. **Some conclusions based on limited data analysis (a single sounding observation) is a major defect and is of little interest to the broader scientific community.**

    **Response,** A total of six Intensive Observation Period (IOP) experiments were conducted in different seasons (non-monsoon and monsoon) in 2014 and 2019, and 240 radio sounding balloons were released, with a total of 226 valid data. We selected experiments under typical weather conditions (clear weather).